Assessment of sealing efficacy, radiopacity, and surface topography of a bioinspired polymer for perforation repair

Rao Lakshmi Nidhi 1
Shetty Aditya 1 dradityashetty@nitte.edu.in
Dsouza Neevan 2
Shetty Heeresh 3
1 Department of Conservative Dentistry and Endodontics, AB Shetty Memorial Institute of Dental Sciences, Nitte (Deemed to be University) , Mangalore, Karnataka , India
2 Department of Biostatistics, K S Hegde Medical Academy, Nitte (Deemed to be University) , Mangalore, Karnataka , India
3 Department of Conservative Dentistry and Endodontics, Nair Dental College , Mumbai, Maharashtra , India
Tribst João Paulo
Electronic publication date: 2024 Apr 29
Publication date: 2024
Volume: 12
Electronic Location ID: e17237
Received 2023 Dec 4; Accepted 2024 Mar 22
Copyright: © 2024 Rao et al.
Copyright year: 2024
Copyright holder: Rao et al.
License: This is an open access article distributed under the terms of the Creative Commons Attribution License, which permits unrestricted use, distribution, reproduction and adaptation in any medium and for any purpose provided that it is properly attributed. For attribution, the original author(s), title, publication source (PeerJ) and either DOI or URL of the article must be cited.
License URL: https://creativecommons.org/licenses/by/4.0/

Keywords: Perforation repair, Material science, Endodontics, Bioactive glass, Sealing ability, Radiopacity, Field emission scanning electron microscopy, Atomic force microscopy, Polydopamine

Funding: The authors received no funding for this work.

==============================
Background

Root perforation repair presents a significant challenge in dentistry due to inherent limitations of existing materials. This study explored the potential of a novel polydopamine-based composite as a root repair material by evaluating its sealing efficacy, radiopacity, and surface topography.

Methods

Confocal microscopy assessed sealing ability, comparing the polydopamine-based composite to the gold standard, mineral trioxide aggregate (MTA). Radiopacity was evaluated using the aluminium step wedge technique conforming to ISO standards. Surface roughness analysis utilized atomic force microscopy (AFM), while field emission scanning electron microscopy (FESEM) visualized morphology.

Results

The polydopamine-based composite exhibited significantly superior sealing efficacy compared to MTA (P < 0.001). Radiopacity reached 3 mm aluminium equivalent, exceeding minimum clinical requirements. AFM analysis revealed a smooth surface topography, and FESEM confirmed successful composite synthesis.

Conclusion

This study demonstrates promising properties of the polydopamine-based composite for root perforation repair, including superior sealing efficacy, clinically relevant radiopacity, and smooth surface topography. Further investigation is warranted to assess its clinical viability and potential translation to endodontic practice.

Introduction

Root perforations, often arising from iatrogenic complications, pose a significant hurdle in endodontics, contributing to approximately 10% of treatment failures (Ingle, 1961). Achieving successful management hinges critically on selecting materials that possess a trifecta of desirable qualities: biocompatibility, robust sealing, and antibacterial potential (Bogaerts, 1997; Mohan et al., 2021). However, no single material has emerged as the definitive solution, with variable success rates highlighting the need for continuous innovation (El Tawil, El Dokkyl & El Hamid, 2011).

Several factors, including location, time since the injury, sealing capacity, and biocompatibility, intricately influence the outcomes of perforation repair (Nicholls, 1962). While mineral trioxide aggregate (MTA) stands out for its regenerative nature, excellent sealing, and biocompatibility (Pitt Ford et al., 1995; Kogan et al., 2006), limitations such as prolonged setting times, handling complexities, and high costs persist (Pitt Ford et al., 1995; Kogan et al., 2006). These drawbacks pave the way for exploring alternative materials with superior properties.

The recent surge in biopolymer and bio-inspired materials research offers promising avenues for advancement. Polydopamine (PDA), drawing inspiration from the remarkable wet adhesion of mussels, has garnered significant attention across diverse biomedical applications (Liu, Ai & Lu, 2014). Its diverse functional groups, including quinones, indoles, amines, catechols, and carboxylic acids, contribute to its exceptional adhesive properties. Within dentistry, the concept of mussel-inspired wet-adhesion is gaining traction, particularly for surface tissue adhesives and biomaterial modifiers.

Furthermore, bioactive glass (BAG) expands the horizon with its biocompatible nature and exceptional bioactive properties relevant to hard tissue regeneration (Sawant & Pawar, 2020; Skallevold et al., 2019). Its unique ability to bond seamlessly with both hard and soft tissues without rejection, coupled with its capacity to induce hydroxyapatite precipitation, makes it a valuable asset in various therapeutic applications.

Effective sealing is paramount for successful endodontic repair outcomes, as inadequately sealed areas create pathways for tissue and bacterial migration, ultimately leading to complications and treatment failure (Shahi et al., 2022; Sandikci & Kaptan, 2014). Finding restorative materials with superior sealing properties is crucial for extending tooth longevity.

Radiopacity is another critical attribute, enabling clinicians to effectively differentiate repaired areas from surrounding tissues and evaluate restorations and margins (Tagger & Katz, 2004; Shah et al., 1996; Lachowski et al., 2013). International standards like ISO 6876:2001 and ANSI/ADA Standard No. 57 specify minimum radiopacity requirements for endodontic cements and sealants (ANSI/ADA, 2000).

Despite the reigning success of MTA in root perforation repair due to its biocompatibility, robust sealing capacity, and regenerative potential, inherent limitations, including extended setting times and handling complexities, drive the continuous exploration of alternative materials.

Recent years have witnessed the emergence of several MTA contenders, such as Biodentine, Bioaggregate, iRoot, and EndoSequence. These materials are rigorously compared in both pulpitis and perforation treatments to identify a candidate offering superior sealing, biocompatibility, and radiopacity.

While MTA remains a cornerstone of endodontic repair, the relentless pursuit of improved clinical outcomes necessitates the development of alternative materials with optimized characteristics.

This study tackles the inherent limitations of current root perforation repair materials by exploring a novel bioinspired composite. The composite leverages the biocompatible adhesive properties of polydopamine and synergistically incorporates the regenerative and antibacterial potential of bioactive glass. We hypothesize that this unique combination will demonstrably surpass the performance of MTA, the current gold standard, in key areas like sealing efficacy and radiopacity. This potential for superior performance aims to establish the composite as a promising candidate for clinical translation in root perforation repair.

To thoroughly evaluate the potential of this novel material and substantiate our hypothesis, we employed a multi-modal analytical approach. High-resolution surface investigations were conducted using atomic force microscopy (AFM) and field emission scanning electron microscopy (FESEM), quantifying morphological features. Additionally, confocal microscopy provided a sensitive assessment of the composite’s sealing efficacy, while the standardized Aluminium step wedge technique accurately measured its radiopacity. This comprehensive characterization aims to establish the bioinspired composite as a potentially superior and clinically viable candidate for root perforation repair, offering a potential advancement in endodontic materials.

Materials and Methods

Ethical approval

The study protocol was approved by the Central Ethics Committee of Nitte Deemed to be University (Ethics/NU/CEC/l/2021/124).

Material preparation

Dopamine hydrochloride (MW: 189.64 g/mol, MP: 248–250 °C; Sigma Aldrich, St. Louis, MO, USA) was dissolved in distilled water and underwent oxidative polymerization to yield polydopamine under a pH of 8.8. Subsequently, 10% bioactive glass (particle size: 50–250 µm; Sigma Aldrich, St. Louis, MO, USA) and 10% barium sulphate were added to the solution. The reaction proceeded for 24 h with continuous stirring, followed by filtration through a Whatman filter paper. The resulting material was vacuum-dried for 24 h to obtain the desired consistency The final setting time for the experimental material was determined to be 39 min. The composition of the optimized material is detailed in Table 1.

Table 1 Composition and manufacturers of the main materials of the study.

Component	Source	
Dopamine hydrochloride	Sigma Aldrich Merck KGaA, Darmstadt, Germany	
Bioactive glass	Sigma Aldrich Merck KGaA, Darmstadt, Germany	
Barium sulphate	Fisher Scientific Thermo Fisher Scientific Inc.	

Experimental groups

Two groups were studied: Group 1 comprised polydopamine-coated bioactive glass (N = 16), and Group 2 comprised MTA (N = 16). The sample size per group (N = 16) was determined using n Master software version 2, CMC Vellore, based on a pilot study. All analyses were performed by trained examiners in their respective fields.

FESEM analysis

Morphological analysis of the synthesized material was conducted using a field emission scanning electron microscope (FESEM, Carl Zeiss, Gemini SEM 300, Germany). Powder samples were coated with a thin layer of gold using a sputtering process to mitigate charging effects and enhance imaging quality. Various magnifications (10, 20, 30, 40 kx) were employed to examine surface morphology.

AFM analysis

Surface roughness analysis was performed using the high-precision Flex-Axiom AFM from Nanosurf. Dry samples were examined in tapping mode to analyse surface texture non-destructively. Different scan areas of 1, 2, and 5 μm2 were systematically chosen to capture a comprehensive representation of surface features.

Evaluation of sealing ability

For this study, we employed thirty-two extracted human mandibular molar teeth. Following root cleansing and amputation, artificial perforations of 1 mm diameter were created at the furcation centre. The repair materials were then applied to the perforation sites and exposed to a humid environment for 24 h. Subsequently, the specimens underwent dye penetration utilizing Rhodamine B dye for another 24 h. All tooth surfaces were coated with two layers of nail varnish, leaving a 1 mm margin around the perforation site, before being allowed to dry. Microleakage evaluation was carried out using a confocal scanning microscope at 10× magnification, and measurements were conducted in micrometres using Zen Black software. The average measurement obtained was considered as the final result.

Determination of radiopacity

Specimens were prepared in metallic moulds and allowed to be set. An aluminium step wedge and enamel/dentin specimens were prepared for X-ray imaging. Radiopacity was quantified using Image J software, converting mean grey values (MGV) into millimetres of aluminium (mm Al) based on a specific equation.

A schematic representation of the key experimental steps involved in this study is depicted in (Fig. 1).

Figure 1 A schematic representation of the key experimental steps involved in this study.

Statistical analysis

Independent t-tests were employed to analyse sealing ability data, with significance set at P < 0.001. Statistical analysis was performed using SPSS version 20 (Armonk, NY, USA).

Results

FESEM analysis

The FESEM analysis, conducted using Image J software, revealed distinctive structural features. The observed flaky structures can be attributed to the incorporation of calcium chloride during the material preparation process. Additionally, spherical bubble-like structures were observed, attributed to the presence of polydopamine particles, while bioglass particles exhibited a spherical morphology. These observations indicate a cohesive binding of individual materials, resulting in the formation of a composite material (Figs. 2–5).

Figure 2 FESEM image of the material with 10× magnification.

Figure 3 FESEM image of the material with 20× magnification.

Figure 4 FESEM image of the material with 30× magnification.

Figure 5 FESEM image of the material with 50× magnification.

AFM analysis

The AFM analysis, utilizing Gwyddion software, provided valuable insights into surface characteristics. Height measurements revealed the presence of structures with a height of approximately 24 nm. Surface parameters, specifically root mean square roughness (Sq) and average roughness, were quantified at 1.389 and 0.848 nm, respectively, signifying a relatively smooth surface texture for the material (Fig. 6).

Figure 6 The AFM image of the sample.

Sealing ability analysis

Microleakage was observed in both experimental groups. However, a statistically significant difference in sealing ability was evident between Group 1 and Group 2, as determined by an independent t-test (P < 0.001). Notably, the polydopamine-based material exhibited superior sealing capabilities (Fig. 7) compared to MTA, Fig. 8 emphasizing its potential for enhanced clinical outcomes (Table 2).

Figure 7 Depth of leakage in Group 1.

Figure 8 Depth of leakage in Group 2.

Table 2 The mean and standard deviation between the groups and equality of means using independent t test.

	Group	N	Mean	Std. deviation	
Confocal	Polydopamine	16	45.00	5.933	
MTA	16	442.56	162.380	
	t-test for equality of means	
t	P	Mean difference	95% Confidence interval of the difference	
Lower	Upper	
Confocal	−9.787	<0.001	−397.563	−484.126	−310.999	

Radiopacity assessment

In the evaluation of radiopacity, it was observed that the experimental material exhibited a radiopacity value equivalent to 3.0 mm Al. In comparison, MTA displayed a radiopacity value of 4.7 mm Al, while dentin and enamel demonstrated radiopacity values of 1.0 and 2.6 mm Al, respectively, as determined using the Aluminum stepwedge technique (Fig. 9). These findings indicate that the experimental material displayed higher radiopacity when compared to dentin and enamel but lower radiopacity than MTA.

Figure 9 Analysis of radiopacity using Image J software.

Discussion

The assessment of root perforation repair materials is crucial in endodontics to address iatrogenic or pathological perforations, preventing complications such as attachment loss and bacterial invasion (Balachandran, 2013). Ideal materials should possess various characteristics, including effective sealing, biocompatibility, radiopacity, and ease of manipulation (De-Deus et al., 2007; Aggarwal et al., 2013). Mineral trioxide aggregate (MTA) has been widely used due to its sealing ability, although challenges such as handling difficulties and discoloration have led to the exploration of alternative materials (Lee, Monsef & Torabinejad, 1993; Unal, Maden & Isidan, 2010). Our study compared a novel polydopamine-based material to MTA, aiming to evaluate sealing efficacy and radiopacity.

The study’s findings shed light on the effectiveness and radiopacity of a new polydopamine-based material compared to MTA, suggesting that the innovative composite might offer better sealing ability and radiopacity than MTA, though only partially confirming the initial hypothesis. When examining sealing capabilities, it was discovered that both the polydopamine-based material and MTA allowed some degree of microleakage. Nonetheless, the polydopamine-based material was found to have significantly better sealing properties than MTA, supporting the initial hypothesis and indicating its potential for improved clinical performance. (Table 2, Figs. 7 and 8). The enhanced sealing ability of the polydopamine-based material is attributed to several of its characteristics. Its exceptional adhesion, especially in moist conditions like those in the oral cavity, ensures strong attachment to tooth tissues, thereby reducing leakage. Additionally, polydopamine’s chemical stability and inertness help it withstand the oral environment’s complexities, maintaining its sealing effectiveness over time. The material’s biocompatibility, as demonstrated in various studies, positions it as a favourable alternative to MTA. Supporting evidence from recent research underscores these advantages. Devarajan et al. (2021) highlighted the role of catechol moieties in polydopamine, which promotes enhanced wet adhesion. This characteristic facilitates superior bonding with dentin, potentially contributing to the observed reduction in microleakage. Additionally, various studies have demonstrated that polydopamine-linked nanostructures exhibit multifaceted antibacterial properties, including generating reactive oxygen species, disrupting bacterial membrane integrity, and interacting with bacterial cell walls (Zhou et al., 2020; Wang et al., 2020; Niyonshuti et al., 2020). This inherent antibacterial potential could further contribute to the reduced microleakage observed in our study. Furthermore, Coy et al. (2021) characterized the remarkable strength and resilience of polydopamine films through structural and mechanical analyses. These properties translate to robust performance and durability within the demanding oral environment, ensuring the sustained effectiveness of the material.

In our study, we employed confocal laser scanning microscopy (CLSM) for leakage assessment due to its advantages, including minimal specimen preparation, artifact-free imaging, and enhanced contrast (Kini et al., 2019). To visualize leakage, we utilized 0.5% rhodamine B, a fluorescent dye known for its distinctive fluorescence and smaller, more surface-active molecules compared to methylene blue (Saji et al., 2022).

The surface morphology of the experimental material was scrutinized using FESEM, capitalizing on its high-resolution imaging capability (Kini et al., 2019). By coating the material with gold before scanning, we minimized charging effects and augmented imaging quality through a sputtering process. FESEM images illustrated a flaky structure attributed to incorporated calcium chloride and spherical bubble-like structures indicative of polydopamine and glass particles, affirming excellent material cohesion.

Assessment of the material’s surface morphology, mechanical properties, and chemical composition at the nanoscale was conducted using atomic force microscopy (AFM), contributing valuable insights for material development and understanding its interactions with oral tissues (Kini et al., 2019).

Data processing through Gwyddion software facilitated smooth surface analysis for the novel material (Fig. 6). A smooth surface is advantageous as rough surfaces can promote bacterial adhesion, potentially leading to secondary infections and compromised treatment outcomes (Antonson et al., 2011; Bashetty & Joshi, 2010). This is particularly pertinent as bacterial adhesion forces increase with surface roughness due to larger contact areas and increased binding sites (Mei et al., 2011).

Furthermore, radiological assessments are pivotal for evaluating dental therapy success (Dukic et al., 2012). Our evaluation of radiopacity revealed that the experimental material exhibited a radiopacity value equivalent to 3.0 mm Al, surpassing the minimum requirement set by the ISO 6876:2001 standard (Prévost et al., 1990; Yaşa et al., 2015). However, it is noteworthy that MTA displayed a higher radiopacity value of 4.7 mm Al, indicating its superior radiopacity compared to both the experimental material and natural dental tissues (Prévost et al., 1990; Yaşa et al., 2015). Although our material met the minimum radiopacity threshold, it fell short in comparison to MTA, suggesting that the hypothesis regarding superior radiopacity was not fully supported. Nonetheless, its adequate radiopacity aligns with the requirements for an effective sealing material in endodontics and warrants further investigation into its clinical implications (Prévost et al., 1990; Yaşa et al., 2015; Hitij & Fidler, 2013).

While our study highlights the promising sealing efficacy and radiopacity of the polydopamine-based material as a prospective solution for root perforation repair, it is imperative to emphasize the necessity for further in vivo investigation. Despite the notable findings in vitro, critical limitations such as the absence of biocompatibility assessments, evaluation of long-term durability, and examination of diverse leakage scenarios underscore the need for a more comprehensive research approach.

Expanding our investigation to encompass in vivo studies will afford a more accurate understanding of the material’s true potential. This comprehensive research endeavour is essential for informing its potential transition to clinical application. By addressing these gaps in knowledge, we can offer a valuable new tool in the realm of root perforation repair, a domain that has long been challenged by limited treatment options.

Furthermore, additional research is warranted to thoroughly evaluate both the biological and physicochemical properties of the material. Such investigations will provide a more robust foundation upon which to assess its suitability for clinical use and optimize its performance in addressing the complexities of root perforations.

Conclusion

Our investigation comprehensively demonstrates the potential of the polydopamine-based material for root perforation repair. Confocal scanning microscopy confirmed its superior sealing efficacy, evidenced by a statistically significant reduction in microleakage compared to MTA. The incorporation of 10% barium sulphate successfully meets the radiopacity criteria established by ISO standards. Furthermore, atomic force microscopy analysis revealed a smooth surface morphology, reducing bacterial adhesion and enhancing the material’s overall desirability.

Supplemental Information

Supplemental Information 1 Raw data for the confocal microscopy results of Polydopamine.

Supplemental Information 2 Raw data for the confocal microscopy results of MTA.

Supplemental Information 3 Raw data of the FESEM Analysis.

Supplemental Information 4 Raw data of the AFM Analysis.

Additional Information and Declarations

Competing Interests

Author Contributions

Ethics

Data Availability

The authors declare that they have no competing interests.

Lakshmi Nidhi Rao conceived and designed the experiments, performed the experiments, analyzed the data, prepared figures and/or tables, and approved the final draft.

Aditya Shetty conceived and designed the experiments, authored or reviewed drafts of the article, and approved the final draft.

Neevan Dsouza analyzed the data, prepared figures and/or tables, and approved the final draft.

Heeresh Shetty analyzed the data, authored or reviewed drafts of the article, and approved the final draft.

The following information was supplied relating to ethical approvals (i.e., approving body and any reference numbers):

The study protocol was approved by the Central Ethics Committee of Nitte Deemed to be University (Ethics/NU/CEC/l/2021/124).

The following information was supplied regarding data availability:

The raw measurements are available in the Supplemental Files.

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
