# Peer review of "Assessment of sealing efficacy, radiopacity, and surface topography of a bioinspired polymer for perforation repair"

_PeerJ, doi:10.7717/peerj.17237_

## Round 0.1 · original submission · Major Revisions

The study in endodontics showcases strengths like a novel approach and commendable study design. However, critical improvements are needed in areas such as methodology details, ethical considerations, and discussion depth. Specific recommendations include adding a schematic view, providing more information on the Evaluation of Sealing Ability, addressing limitations, proposing future directions, and enhancing figure clarity. Given these concerns, I am requesting major revisions, as the manuscript currently falls short of the required scientific rigor and clarity for publication in PeerJ.

Reviewer 1 ·

Basic reporting

- The English should be revised to enhance clarity for a global audience, ensuring that it is easily comprehensible across various linguistic backgrounds. For instance, the paragraph starting at line 63 is somewhat unclear and could be rewritten to improve its flow and readability. In addition, some sentences are quite long and could be simplified.
- The use of 'evaluating' in the title might imply ongoing development. If the authors deem it relevant, I would suggest replacing this verb to better reflect the study's completed analysis.
- In the abstract the background description is super extensive and could be better summarized, on the other hand, the materials and methods were described very succinctly and could be explored much better.
- The introduction is well-structured and provides a good background with literature references. However, clarity could be improved in certain sections. For instance, the paragraph beginning at line 63 might benefit from additional exploration, particularly if the authors elaborate on the mechanism of action that renders these materials noteworthy. In addition, it would be interesting if the authors highlighted the “gap in knowledge” to the reader. No hypotheses was provided.
- The article's discussion effectively integrates the results, however I suggest that the discussion be reviewed and updated, it should be more robust, with a greater grounding in recent literature that reflects the current science on the topic (The most recent reference is from 2019, only 1).
- Line 165 ‘‘Mineral trioxide aggregate (MTA) has exhibited a high success rate in furcation perforation repair, attributed in part to its extended setting time, reducing material shrinkage after setting, and its hydrophilic nature, which maintains sealing ability in wet environments (20).Despite MTA's favourable properties, challenges such as handling difficulties, extended setting time, and potential tooth discoloration have prompted the development of alternative materials with improved characteristics.’’ It seems that we have a contradiction, after all, the extended setting time is a positive or negative point of the MTA?

Experimental design

- I suggest the authors to add a ‘‘table 1’’ with the composition and manufacturers of the main materials used in this study.
- Thirty-two extracted human mandibular molar teeth were used, however there are no information regarding ethical committee approval.
- There are no specification regarding the samples that were used to the Field Emission Scanning Electron Microscopy (FESEM) and Atomic Force Microscopy (AFM) analysis. Sample size? Dimensions?
- What magnification was used for the FESEM analysis?
- Line 110, information about the experimental group should be at the beginning of the materials and methods section, along with the sample size and sample dimensions.
- I recommend enhancing the description of the methodology for sealing ability analysis. It would be beneficial to specify the magnification used for confocal analyses. Also, the process, including whether the teeth were immersed in 0.2% rhodamine B before being sectioned for confocal microscope examination, needs clarification. Furthermore, detailing how the dye penetration was assessed, such as the software used and the unit of measurement, would significantly contribute to the clarity and reproducibility of the study.
-The authors do not mention whether the analyzes were carried out by trained or calibrated examiners.
-Table 2 is unnecessary. I suggest the authors to add the information contained in table 2 in the table 1.
-The authors should insert a paragraph in the discussion about the clinical relevance and the limitations of this study.

Validity of the findings

no comment

Annotated reviews are not available for download in order to protect the identity of reviewers who chose to remain anonymous.

Reviewer 2 ·

Basic reporting

This manuscript investigated the Sealing Efficacy, Radiopacity, and Surface Topography of a Bioinspired Polymer for Perforation Repair. The manuscript is well designed and well written. Additionally, I did not see any methodological problems. I commend the authors on their efforts in preparing the manuscript. However, I would like the authors to address the following points;
1) Introduction: The introduction part is well written. The problem is significant in the field of endodontics. The novelty of the research is stated in the introduction part.
2) Methods: I think authors could add a schematic view of the experimental set up.
3) Methods: Could you please provide detail for the Evaluation of Sealing Ability? How many images did you scan? How did you measure the extent of microlekage? Did you take average or did you measure the longest one?
4) Discussion: What are the limitations of this study?
5) Discussion: Authors could add a note about future directions of this study.
6) For Figure 1, authors could add A, B, C or D on small images and give the magnifications in figure caption.

Experimental design

I've stated above.

Validity of the findings

I've stated above.

Additional comments

I've stated above.

·

Basic reporting

Highlights:

• The abstract provides a concise summary of the research, delivering a clear overview of the key findings.
• The introduction effectively establishes the context and justification for the study, emphasizing the need for MTA as a perforation sealing material.
• Detailed information is provided on the composition and preparation of the new material.

Recommended Corrections:
• Abstract: Condense the background section, eliminate repetitions of testing methods, and enhance clarity on the study design. Include all testing parameters in the methodology.
The abstract provides a concise overview of the research, effectively summarizing the findings. However, to enhance its clarity and effectiveness, the background section should be condensed. Additionally, removing repetitions of testing methods and incorporating all testing parameters into the methodology will streamline the abstract and avoid redundancy.

• Introduction: While discussing MTA, acknowledge other promising materials. Explicitly state the need for testing methodologies like Atomic Force Microscopy (AFM) and Field Emission Scanning Electron Microscopy (FESEM).
The introduction successfully highlights the significance of MTA as a perforation sealing material. To further enrich the discussion, it is advisable to acknowledge other promising materials explored in various research studies. This not only broadens the context but also positions the current study within the larger landscape of perforation sealing materials. Furthermore, explicitly stating the need for testing methodologies like Atomic Force Microscopy (AFM) and Field Emission Scanning Electron Microscopy (FESEM) will enhance the justification for their inclusion.

Experimental design

Highlights:

• Utilization of AFM and FESEM for thorough material characterization.
• Methodology outlines micro leakage assessment, sample size determination, and effect size calculation.
• Inclusion and exclusion criteria are mentioned for samples, ensuring study integrity.

Recommended Corrections:

• In the material preparation section, specify the form of the resultant material before AFM and FESEM.
While the experimental design effectively employs Atomic Force Microscopy (AFM) and Field Emission Scanning Electron Microscopy (FESEM) for material characterization, additional clarity is needed in the material preparation section. Specifically, specifying the form of the resultant material before subjecting it to AFM and FESEM will provide essential insights. Understanding whether the material is in powder form and, if so, the components it is mixed with will enhance the comprehensibility of subsequent analyses.

• Provide details on sample preparation, including mixing, sample description, and dimensions of perforation defects.
The methodology section outlines micro leakage assessment but lacks specific details on sample preparation. It would be valuable to elaborate on the mixing process, describe the resultant sample, and provide dimensions of the perforation defects. This additional information is essential for replicability and a more thorough understanding of the experimental setup.


• Include information on root canal exploration status and novel material handling, such as setting time.
The study would benefit from including details regarding the status of root canal exploration. This information is pertinent to understanding the baseline conditions of the samples. Additionally, novel material handling information, including setting time, should be provided to offer a comprehensive view of the experimental parameters.

• Specify sample preparation details before dye leakage assessment, addressing interim restorations, surface coatings, and assessment parameters.
Further details on the preparation of samples before subjecting them to dye leakage methodology are essential for a complete understanding of the experimental process. This includes information on whether access interim restorations were placed, any coatings (such as nail varnishes) on surfaces other than the area of interest, and a clear indication of whether the leakage assessment was conducted from the internal or external surface of the perforations.

Validity of the findings

Highlights:

• The use of the novel material is promising for the field of dentistry.
• Positive sealing ability demonstrated by leakage measurements and microscopic analysis.
• The discussion elaborates on the results considerably, connecting them to existing literature and highlighting potential implications.

Recommended Corrections:

• In the discussion, avoid repeating statements about perforations and MTA disadvantages. Elaborate more on result-oriented discussion.
While the discussion effectively highlights the promising nature of the novel material and its positive sealing ability, there is a tendency to repeat statements regarding perforations and MTA disadvantages. To enhance the quality of the discussion, it is advisable to minimize repetitions and focus on a more result-oriented analysis. Delving deeper into the implications of the results, potential applications in dentistry, and any limitations encountered during the study will contribute to a more robust and insightful discussion.


• Provide a reference for the statement regarding material structure characteristics.
The statement referencing the material structure characteristics is intriguing, but to strengthen its validity, it is crucial to provide a supporting reference. This will allow readers to explore the source of the information and gain confidence in the interpretation of the material's structural features.

• Clarify if statements about resin composites on the clinical crown portion apply to the perforation repair site.
A critical point raised in my review is the applicability of statements about resin composites on the clinical crown portion to the perforation repair site. Justification for the relevance of these statements to the specific context of perforation repair, particularly considering the challenges in achieving a smooth surface on the tissue side, needs to be provided. This clarification will strengthen the connection between the referenced literature and the current study.

• Correct references as per the suggestions provided, address typographic errors, and label figures for better understanding.
The recommendations related to references include addressing formatting issues, such as following a consistent pattern for author names and checking for abbreviations of journal names. Additionally, addressing typographic errors, ensuring clarity in figure labelling, and providing explanations below tables regarding the units of values will contribute to a more polished and professional presentation.


Some of the noted errors

• Ref no 3: List all authors' names up to the sixth, followed by "et al" if there are more than six authors.
• Ref no 8: Eliminate the conjunction "and" from the list of authors' names.
• Ref no. 10: Verify and rectify any abbreviations used for the journal name.
• Ref No 13: Replace any abbreviations employed for the journal name with the complete name.
• Ref No 14: Confirm and correct any abbreviations used for the journal name.
• Ref No 18: Specify all authors' names up to the sixth, followed by "et al" if there are more than six authors.
• Ref No 22 and 23: Ensure consistency in the format of the year and page numbers.
• Ref No. 28: Address any typographic errors occurring after the reference.
• Duplications in figure and table headers.
• Ensure that images are properly labelled to enhance understanding of structures and interfaces.
• Correct any typographic errors present in Table 1.
• Provide an explanation beneath the table clarifying the units of values used

Additional comments

The study is novel with a good study design, requiring corrections in the mentioned points and uniform referencing patterns for journal name abbreviations.
My comments highlight the overall novelty of the study and its commendable study design. However, to elevate the quality of the manuscript, it is crucial to address the specific points mentioned in my review. Achieving uniformity in referencing patterns for journal name abbreviations will enhance the professionalism and consistency of the manuscript.

---

## Round 0.2 · accepted · Accept

Dear author,

I am delighted to inform you that your manuscript titled "Assessment of sealing efficacy, radiopacity, and surface topography of a Bioinspired polymer for perforation repair" has been accepted for publication in PeerJ. Based on the thorough revisions and the quality of the current version, I believe that the manuscript is ready for publication.

Kind regards,
Dr. Tribst JPM

Reviewer 1 ·

Basic reporting

The authors answered all the requests of the reviewers.

Experimental design

The authors answered all the requests of the reviewers.

Validity of the findings

The authors answered all the requests of the reviewers.

Additional comments

Overall, the manuscript is improved and presents an interesting topic with results that can provide significance in clinical application. The authors resolved the elucidated concerns, made all the corrections, and addressed all indicated issues.

Reviewer 2 ·

Basic reporting

After carefully reviewing the manuscript and the revisions made by the authors, I am pleased to report that I find the revisions satisfactory. The authors have diligently addressed all the concerns raised during the previous review process, resulting in a significantly improved manuscript. Their responses were thorough and reflective, demonstrating a clear commitment to enhancing the quality and clarity of their work. Based on the revisions made, I am confident in recommending acceptance of the manuscript for publication. I commend the authors for their diligence and responsiveness throughout this process.

Experimental design

.

Validity of the findings

.

·

Basic reporting

All the suggested changes have been done.

Experimental design

Majority of the corrections suggested have been rectified, however in the dye leakage section the coating of nail varnish have been done after the samples were subjected to dye leakage, which is rather questionable. kindly rectify

Validity of the findings

Suggested changes were implemented, however the repetitions of figure legends can be removed

Additional comments

The study has novelty in terms of new material testing for the perforation. The article can be accepted for publication after correcting the suggested changes.